# Mediating artificial intelligence developments through negative and positive incentives

**The Anh Han** [1]*, **Luís Moniz Pereira** [2], **Tom Lenaerts** [3,4], **Francisco C. Santos** [5]

**1** School of Computing and Digital Technologies, Teesside University, Middlesbrough, United Kingdom,
**2** NOVA Laboratory for Computer Science and Informatics (NOVA LINCS), Universidade Nova de Lisboa,
Caparica, Portugal, **3** Machine Learning Group, Université Libre de Bruxelles, Boulevard du Triomphe,
Brussels, Belgium, **4** Artificial Intelligence Lab, Vrije Universiteit Brussel, Brussels, Belgium, **5** INESC-ID and
Instituto Superior Tecnico, Universidade de Lisboa, Porto, Salvo, Portugal

\* t.han@tees.ac.uk

## Abstract

The field of Artificial Intelligence (AI) is going through a period of great expectations, introducing a certain level of anxiety in research, business and also policy. This anxiety is further energised by an AI race narrative that makes people believe they might be missing out. Whether real or not, a belief in this narrative may be detrimental as some stake-holders will feel obliged to cut corners on safety precautions, or ignore societal consequences just to "win". Starting from a baseline model that describes a broad class of technology races where winners draw a significant benefit compared to others (such as AI advances, patent race, pharmaceutical technologies), we investigate here how positive (rewards) and negative (punishments) incentives may beneficially influence the outcomes. We uncover conditions in which punishment is either capable of reducing the development speed of unsafe participants or has the capacity to reduce innovation through over-regulation. Alternatively, we show that, in several scenarios, rewarding those that follow safety measures may increase the development speed while ensuring safe choices. Moreover, in the latter regimes, rewards do not suffer from the issue of over-regulation as is the case for punishment. Overall, our findings provide valuable insights into the nature and kinds of regulatory actions most suitable to improve safety compliance in the contexts of both smooth and sudden technological shifts.

## Introduction

With the current business and governmental anxiety about AI and the promises made about the impact of AI technology, there is a risk for stake-holders to cut corners, preferring rapid deployment of their AI technology over an adherence to safety and ethical procedures, or a willingness to examine their societal impact [1–3].

Agreements and regulations for safety and ethics can be enacted by involved parties so as to ensure their compliance concerning mutually adopted standards and norms [4]. However, experience with a spate of international treaties, like those of climate change, timber, and fisheries agreements [5–7] has shown, the autonomy and sovereignty of the parties involved will

---

**Data Availability Statement:** All relevant data are within the manuscript and its Supporting information files.

**Funding:** T.A.H., L.M.P. and T.L. have been supported by Future of Life Institute grant RFP2-

154. T.A.H. is also supported by a Leverhulme Research Fellowship (RF-2020-603/9). L.M.P. is also supported by NOVA LINCS (UIDB/04516/2020) with the financial support of FCT-Fundação para a Ciência e a Tecnologia, Portugal, through national funds. F.C.S. acknowledges support from FCT Portugal (grants UIDB/50021/2020, PTDC/MAT-APL/6804/2020, and PTDC/CCI-INF/7366/2020). T.L. and F.C.S. acknowledge the support by TAILOR, a project funded by EU Horizon 2020 research and innovation programme under GA No 952215. T.L. acknowledges support by the FuturICT2.0 (www.futurict2.eu) project funded by the FLAG-ERA JTC 2016.

**Competing interests:** The authors have declared that no competing interests exist.

make monitoring and compliance enforcement difficult (if not impossible). Therefore, for all to enjoy the benefits provided by safe, ethical and trustworthy AI, it is crucial to design and impose appropriate incentivising strategies in order to ensure mutual benefits and safety-compliance from all sides involved. Given these concerns, many calls for developing efficient forms of regulation have been made [2, 8, 9]. Despite a number of proposals and debates on how to avert, regulate, or mediate a race for technological supremacy [2, 4, 8–12], few formal modelling studies were proposed [1, 13]. The goal of the this work is to further bridge this crucial gap.

We aim to understand how different forms of incentives can be efficiently used to influence safety decision making within a development race for domain supremacy through AI (DSAI), resorting to population dynamics and Evolutionary Game Theory (EGT) [14–16]. Although AI development is used here to frame the model and to discuss the results, both model and conclusions may easily be adopted for other technology races, especially where a winner-takes-all situation occurs [17–19].

We posit that it requires time to reach DSAI, modelling this by a number of development steps or technological advancement rounds [13]. In each round the development teams (or players) need to choose between one of two strategic options: to follow safety precautions (the SAFE action) or ignore safety precautions (the UNSAFE action). Because it takes more time and more effort to comply with precautionary requirements, playing SAFE is not just costlier, but implies slower development speed too, compared to playing UNSAFE. We consequently assume that to play SAFE involves paying a cost $c > 0$, while playing UNSAFE costs nothing ($c = 0$). Moreover, the development speed of playing UNSAFE is $s > 1$ whilst the speed of playing SAFE is normalised to $s = 1$. The interaction is iterated until one or more teams establish DSAI, which occurs probabilistically, i.e. the model assumes, upon completion of each round, that there is a probability $\omega$ that another development round is required to reach DSAI—which results in an average number $W = (1 - \omega)^{-1}$ of rounds per competition/race [16]. We thus do not make any assumption about the time required to reach DSAI in a given domain. Yet once the race ends, a large benefit or prize $B$ is acquired that is shared amongst those reaching the target simultaneously.

The DSAI model further assumes that a development setback or disaster might occur, with a probability assumed to increase with the number of occasions the safety requirements have been omitted by the winning team(s) at each round. Although many potential AI disaster scenarios have been sketched [1, 20], the uncertainties in accurately predicting these outcomes have been shown to be high. When such a disaster occurs, the risk-taking participant loses all its accumulated benefits, which is denoted by $p_r$, the risk probability of such a disaster occurring when no safety precaution is followed (see Materials and methods section for further details).

As shown in [13], when the time-scale of reaching the target is short, such that the average benefit over all the development rounds, i.e. $B/W$, is significantly larger compared to the intermediate benefit obtained in every round, i.e. $b$, there is a large parameter space where societal interest is in conflict with the personal one: unsafe behaviour is dominant despite the fact that safe development would lead to a greater social welfare (see Methods for more details). From a regulatory perspective, only that region requires additional measures that ensure or enhance safe and globally beneficial outcomes, avoiding any potential disaster. Large-scale surveys and expert analysis of the beliefs and predictions about the progress in AI, indicate that the perceived time-scale for supremacy across domains through AI as well as regions is highly diverse [21, 22]. Also note that despite focusing on DSAI in this paper, the proposed model is generally applicable to any kind of long-term competing situations such as technological innovation development and patent racing where there is a significant advantage (i.e. large $B$) to be

achieved by reaching an important target first [17–19]. Other domains include pharmaceutical development where firms could try to cut corners by not following safe clinical trial protocols in an effort to be the first to develop a pharmaceutical produce (i.e. a cure for cancer), in order to take the highest possible share of the market benefit [23]; Besides tremendous economic advantage, a winner of a vaccine race such as for Covid-19 treatment, can also gain significant political and reputation influence [24].

In this paper, we explore whether and how incentives such as reward and punishment can help in avoiding disasters and generate a wide benefit of AI-based solutions. Namely, players can attempt to prevent others from moving as fast as they want (i.e., an elementary form of punishment of wrong-doers) or help others to speed up their development (rewarding right-doers), at a given cost. Slowing down unsafe participants can be obtained by reporting misconduct to authorities and media, or by refusal to share and collaborate with companies not following the same deontological principles. Similarly, rewards can correspond to support, exchange of knowledge, staff, etc. of safety conscious participants. Note that reasons for intervening with the development speed of competitors may also be nefarious, e.g. cyber-attacks, in order to get a speed advantage. The current work only considers interventions by safe players as a result of the unsafe behaviour of co-players. We show that both negative and positive incentives can be efficient and naturally self-organize (even when costly). However, we also show that such incentives should be carefully introduced, as they can have negative effects otherwise. To this end, we identify the conditions under which positive and negative incentives are conducive to desired collective outcomes.

## Materials and methods

### DSAIR model definition

Let us depart from the innovation race or domain supremacy through AI race (DSAIR) model developed in [13]. We adopt a two-player repeated game, consisting of, on average, $W$ rounds. At each development round, players can collect benefits from their intermediate AI products, subject to whether they choose playing SAFE or UNSAFE. By assuming some fixed benefit, $b$, resulting from the AI market, the teams share this benefit in proportion to their development speed. Hence, for every round of the race, we can write, with respect to the row player $i$, a payoff matrix denoted by $\Pi$, where each entry is represented by $\Pi_{ij}$ (with $j$ corresponding to a column), as follows

$$
\Pi = \begin{matrix} & \begin{matrix} SAFE & \quad UNSAFE \end{matrix} \\ \begin{matrix} SAFE \\ UNSAFE \end{matrix} & \begin{pmatrix} -c + \frac{b}{2} & -c + \frac{b}{s+1} \\ \frac{sb}{s+1} & \frac{b}{2} \end{pmatrix} \end{matrix} \tag{1}
$$

The payoff matrix can be explained as follows. First of all, whenever two SAFE players interact, each will pay the cost $c$ and share the resulting benefit $b$. Differently, when two UNSAFE players interact, each will share the benefit $b$ without having to pay $c$. When a SAFE player interacts with an UNSAFE player, the SAFE one pays a cost $c$ and receives a (smaller) part $b/(s + 1)$ of the benefit $b$, while the UNSAFE one obtains the larger part $sb/(s + 1)$ without having to pay $c$. Note that $\Pi$ is a simplification of the matrix defined in [13] since it was shown that the parameters defined here are sufficient to explain the results in the current time-scale.

We will analyse evolutionary outcomes of safety behaviour within a well-mixed, finite population consisting of $Z$ players, who repeatedly interact with each other in the AI development process. They will adopt one of the following two strategies [13]:

- **AS**: always complies with safety precaution, playing SAFE in all the rounds.

- **AU**: never complies with safety precaution, playing UNSAFE in all the rounds.

Recall that $B$ stands for the big prize shared by players winning a race (together), while $s$ and $p_r$ denote the speed earned by playing UNSAFE (compared to the speed of SAFE being normalised to $1$, $s > 1$) and the probability that AI disaster occurring due to such unsafe behaviour being adopted in all rounds of the race. Thus, the payoff matrix defining averaged payoffs for AU vs AS is given by

$$
\begin{array}{cc}
 & \begin{array}{cc} AS & \qquad\quad AU \end{array} \\
\begin{array}{c} AS \\ AU \end{array} &
\begin{pmatrix}
\frac{B}{2W} + \Pi_{11} & \Pi_{12} \\
p\left(\frac{sB}{W} + \Pi_{21}\right) & p\left(\frac{sB}{2W} + \Pi_{22}\right)
\end{pmatrix}
\end{array}
\tag{2}
$$

where, solely with the purpose of presentation, we denote $p = 1 - p_r$.

As was shown in [13] by considering when AU is risk-dominant against AS, three different regions can be identified in the parameter space $s$-$p_r$ (see Fig 1, with more details being

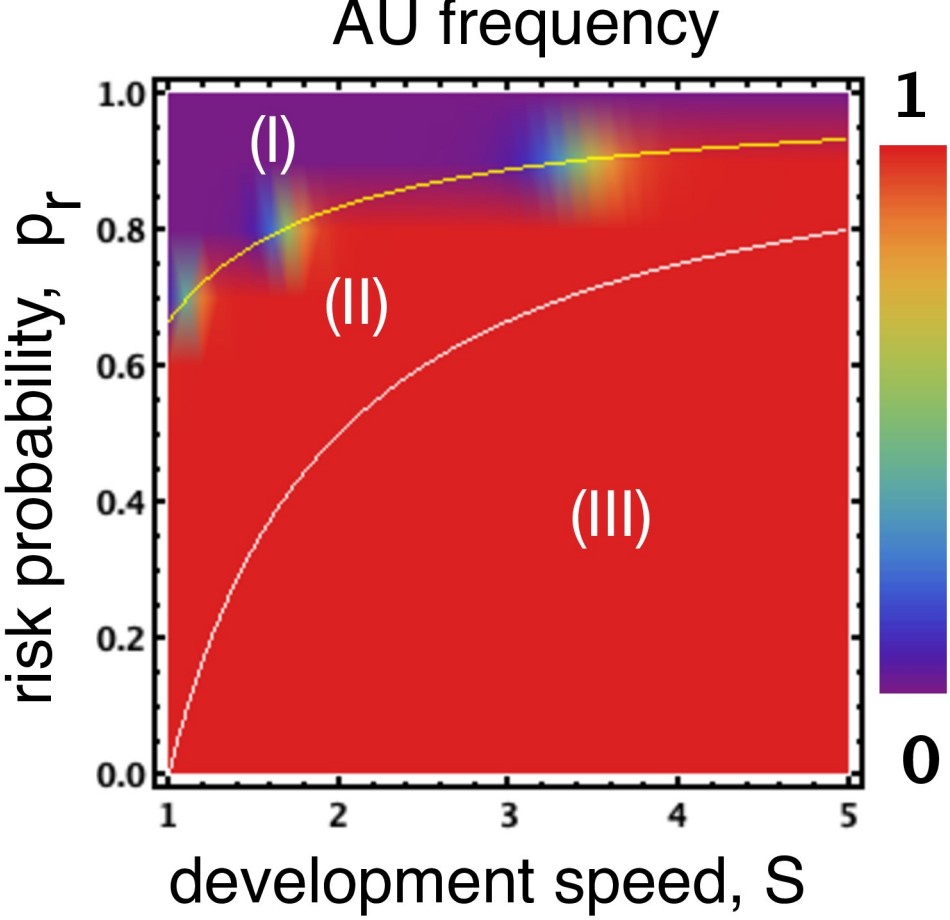

**Fig 1. Frequency of AU in a population of AU and AS.** Region (**II**): The two solid lines inside the plots delineate the boundaries $p_r \in [1 - 1/s,\ 1 - 1/(3s)]$ where safety compliance is the preferred collective outcome yet evolution selects unsafe development. Regions (**I**) and (**III**) display where safe (respectively, unsafe) development is not only the preferred collective outcome but also the one selected by evolution. Parameters: $b = 4$, $c = 1$, $W = 100$, $B = 10^4$, $\beta = 0.01$, $Z = 100$.

provided in SI): (**I**) when $p_r > 1 - \frac{1}{3s}$, AU is risk-dominated by AS: safety compliance is both the preferred collective outcome and selected by evolution; (**II**) when $1 - \frac{1}{3s} > p_r > 1 - \frac{1}{s}$: even though it is more desirable to ensure safety compliance as the collective outcome, social learning dynamics would lead the population to the state wherein the safety precaution is mostly ignored; (**III**) when $p_r < 1 - \frac{1}{s}$ (AU is risk-dominant against AS), then unsafe development is both preferred collectively and selected by social learning dynamics.

That is, only region (**II**) in Fig 1 requires regulatory actions such as incentives to improve the desired safety behaviour. The intuition is that, those who completely ignore safety precautions can always achieve the big prize *B* when playing against safe participants. The two other regions, i.e. region **I** and region **III** in Fig 1, do not suffer from a dilemma between individual and group benefits as is the case for region **II**. Whereas in region **I** safe development is preferred due to excessively high risks, region **III** prefers unsafe, risk taking behaviour, both from an individual and societal perspective, due to low levels of risk.

It is worthy of note that adding a conditional strategy (that, for instance, plays SAFE in the first round and thereafter adopts the same move its co-player used on the previous round) does not influence the dynamics or improve safe outcomes (see details in SI). This is contrary to the prevalent models of direct reciprocity in the repeated social dilemmas context [16, 25, 26]. Therefore, additional measures need to be put in place for driving the race dynamics towards a more beneficial outcome. To this end, we came to explore in this work the effects of negative (sanctions) and positive (rewards) incentives.

## Punishment and reward in innovation races

Given the DSAIR model one can now introduce incentives that affect the development speed of the players. These incentives reduce or increase the speed of development of a player as this is the key factor in gaining *b*, the intermediate benefit in each round, as well as *B*, the big prize of winning the race once the game ends [13]. While there are many ways to incorporate them, we assume here a minimal model where the effect on speed is constant and fixed over time, hence not cumulative with the number of unsafe or safe actions of the co-player. Given this constant assumption, a negative incentive reduces the speed of a co-player taking an UNSAFE action to a lower but constant speed-level. Similarly, a positive incentive increases the speed of a co-player that took a safe action to a fixed higher speed level. In both cases these incentives are attributed in the next round, after observing the UNSAFE or SAFE action respectively. Moreover, both positive and negative incentives are considered to be costly, meaning that the strategy that awards them will reduce its own speed by providing the incentive. Given these assumptions the following two strategies are studied in relation to the AS and AU strategies defined earlier:

- A strategy PS that always plays SAFE but will sanction the co-player after she has played UNSAFE in the previous round. The punishment by PS imposes a reduction $s_\beta$ on the opponent's speed as well as a reduction $s_\alpha$ on her own speed (see Fig 2, orange line/area).

- A strategy RS that always chooses the SAFE action and will reward a SAFE action of a co-player by increasing her speed with $s_\beta$ while paying a cost $s_\alpha$ on her own speed (see Fig 2, blue line/area).

The analysis performed in the Results section aims to show whether having PS or/and RS in the population leads to more societal welfare in the region (**II**), where there is a conflict between individual and societal interests. The methods used in this analysis are discussed in the next section.

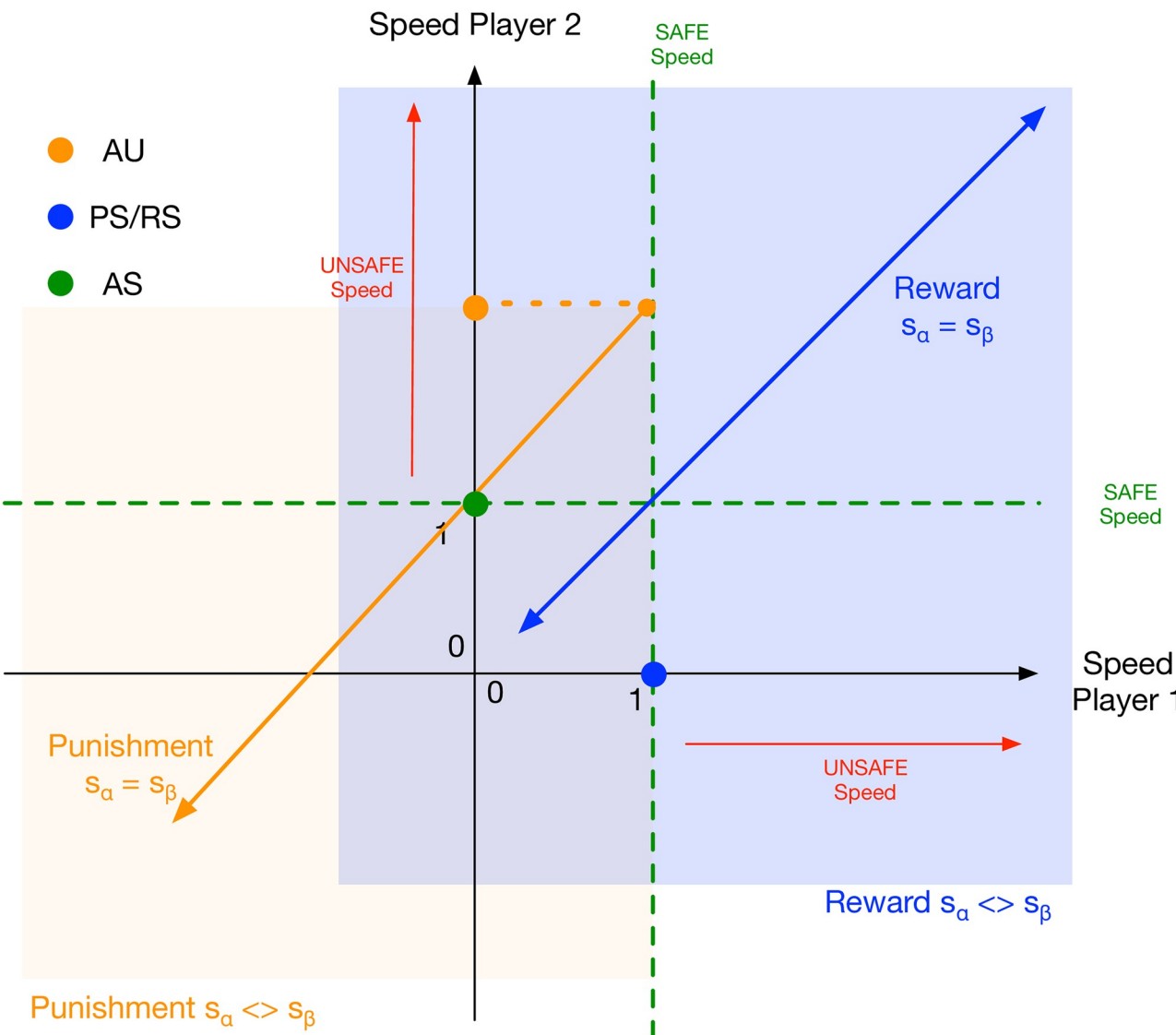

**Fig 2. Effect of positive and negative incentives on players' speed.** On the one hand, when player 1 is of type PS (blue circle on x-axis), i.e. sanctioning unsafe actions, it reduces the future speed of player 2 when she is of type AU (orange circle on the y-axis), while paying a speed cost, possibly equivalent to the reduction in speed that the AU player is experiencing (orange line). In general the reduction of speeds of player 1 and 2 fall into the area marked by the orange rectangle (it is referred in the main text as orange area). On the other hand, when player 1 is of type RS (blue circle on x-axis), i.e. rewarding safe actions, it increases the speed of player 2 (green circle on y-axis), while paying a speed cost that reduces the RS player's speed. Differently from before, the speed effect is in opposing directions for the two players (hence, the blue line is bidirectional). The blue rectangle (referred in the main text as blue area) marks the area of the speed of player 1 and player 2. In the analysis in the paper, first the case of equal speed effects is considered (lines) before analysing different speed effects (rectangles) between both players.

## Evolutionary dynamics for finite populations

We employ EGT methods for finite populations [16, 27, 28], whether in the analytical or numerical results obtained here. Within such a setting, the players' payoffs stand for their *fitness* or social *success*, and social learning shapes the evolutionary dynamics, according to which the most successful players will more often tend to be imitated by other players. Social learning is herein modeled utilising the so-called pairwise comparison rule [27], assuming that

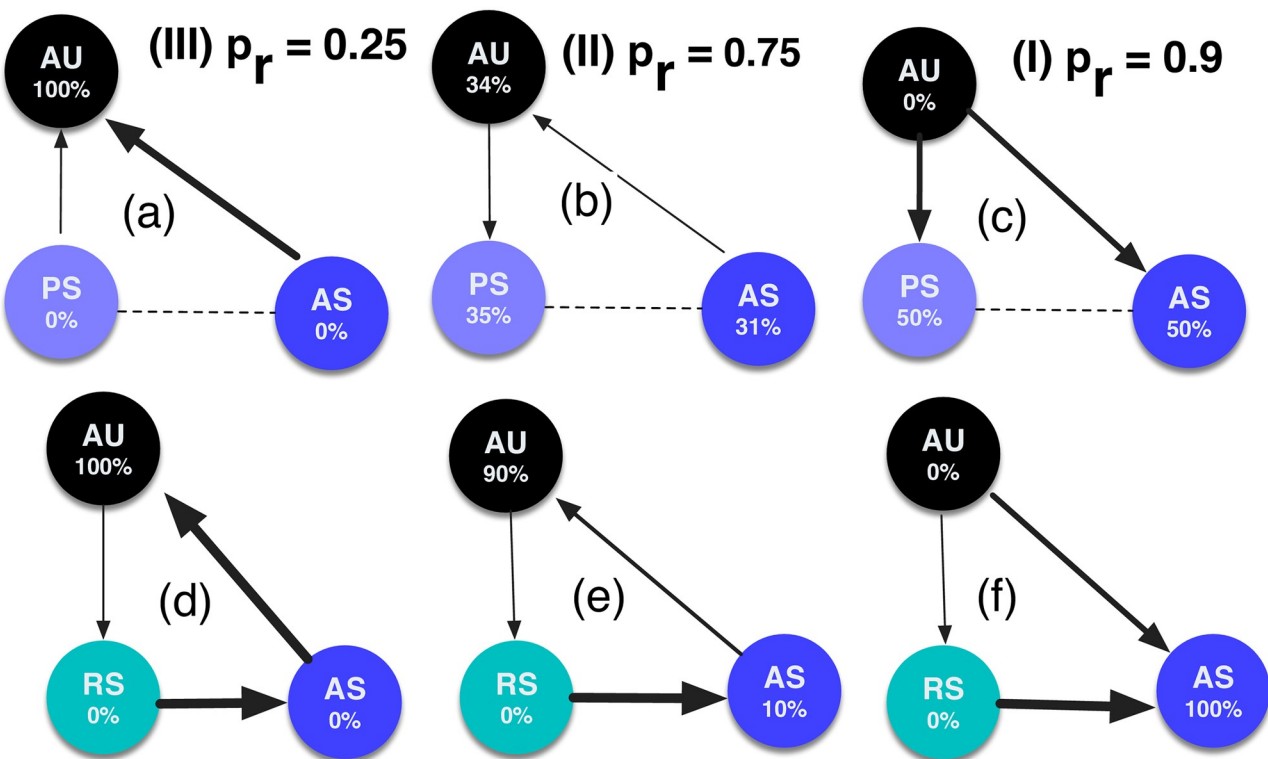

**Fig 3. Transitions and stationary distributions in a population of three strategies AU, AS, with either PS (top row) or RS (bottom row), for three regions.** Only stronger transitions are shown for clarity. Dashed lines denote neutral transitions. Parameters: $s_\alpha = s_\beta = 1.0$, $c = 1$, $b = 4$, $W = 100$, $B = 10000$, $\beta = 0.01$, $Z = 100$.

a player $A$ with fitness $f_A$ adopts the strategy of another player $B$ with fitness $f_B$ with probability assigned by the Fermi function, $P_{A,B} = \left(1 + e^{-\beta(f_B - f_A)}\right)^{-1}$, where $\beta$ conveniently describes the intensity of selection. The long-term frequency of each and every strategy in a population where several of them are in co-presence, can be computed simply by calculating the stationary distribution of a Markov chain whose states represent those strategies. In the absence of behavioural exploration or mutations, end states of evolution inevitably are monomorphic. That is, whenever such a state is reached, it cannot be escaped via imitation. Thus, we further assume that, with some mutation probability, an agent can freely explore its behavioural space (in our case, consisting of two actions, SAFE and UNSAFE), randomly adopts an action therein. At the limit of a small mutation probability, the population consists of at most two strategies at any time. Consequently, the social dynamics can be described using a Markov Chain, where its state represents a monomorphic population and its transition probabilities are given by the fixation probability of a single mutant [29, 30]. The Markov Chain's stationary distribution describes the time average the population spends in each of the monomorphic end states. Below we described the step-by-step details how the stationary distribution is calculated (some examples of fixation probabilities and stationary distributions in a population of three strategies AS, AU and PS or RS can already be seen in Fig 3).

Denote by $\pi_{X,Y}$ the payoff a strategist $X$ obtains in a pairwise interaction with strategist $Y$ (defined in the payoff matrices). Suppose there exist at most two strategies in the population, say, $k$ agents using strategy A ($0 \leq k \leq Z$) and ($Z - k$) agents using strategies B. Thus, the

(average) payoff of the agent that uses A and B can be written as follows, respectively,

$$
\Pi_A(k) = \frac{(k-1)\pi_{A,A} + (Z-k)\pi_{A,B}}{Z-1},
$$
$$
\Pi_B(k) = \frac{k\pi_{B,A} + (Z-k-1)\pi_{B,B}}{Z-1}. \tag{3}
$$

Now, in each time step, the probability of change by ±1 of a number of $k$ agents using strategy A can be specified as [27]

$$
T^{\pm}(k) = \frac{Z-k}{Z}\frac{k}{Z}\left[1 + e^{\mp\beta[\Pi_A(k)-\Pi_B(k)]}\right]^{-1}. \tag{4}
$$

The fixation probability of a single mutant adopting A, in a population of $(Z-1)$ agents adopting B, is specified by [27, 30]

$$
\rho_{B,A} = \left(1 + \sum_{i=1}^{Z-1}\prod_{j=1}^{i}\frac{T^-(j)}{T^+(j)}\right)^{-1}. \tag{5}
$$

When considering a set $\{1, \ldots, s\}$ of distinct strategies, these fixation probabilities determine the Markov Chain transition matrix $M = \{T_{ij}\}_{i,j=1}^{s}$, with $T_{ij,j\neq i} = \rho_{ji}/(s-1)$ and $T_{ii} = 1 - \sum_{j=1,j\neq i}^{s} T_{ij}$. The normalized eigenvector of the transposed of $M$ associated with the eigenvalue 1 provides the above described stationary distribution [29], which defines the relative time the population spends while adopting each of the strategies.

**Risk-dominance.** An important approach for comparing two strategies A and B is that of in which direction the transition is stronger or more probable, that of an A mutant fixating in a population of agents employing B, $\rho_{B,A}$, or that of a B mutant fixating in the population of agents employing A, $\rho_{A,B}$. In the limit, for large population size (large $Z$), this condition can be simplified to [16]

$$
\pi_{A,A} + \pi_{A,B} > \pi_{B,A} + \pi_{B,B}. \tag{6}
$$

## Results

### Negative incentives are a double-edged sword

As explained in Methods PS reduces the speed of an AU player from $s$ to $s - s_\beta$, while reducing its own speed from 1 (since it plays always SAFE) to $1 - s_\alpha$. Hence one can define $s' = 1 - s_\alpha$ as the new speed for PS and $s'' = s - s_\beta$ as the new speed for AU. Depending on the values of $s_\alpha$ and $s_\beta$, these speeds may also be zero or even negative, which represent situations where no progress is being made or where punishment even destroys existing development, respectively. In the following we consider these situations in two different ways. First, a theoretical analysis is performed for the situation where $s_\beta = s_\alpha$. Second, this assumption is relaxed and a numerical study of the generalised case is provided.

There are two scenarios to consider when $s_\beta = s_\alpha$: (i) when $s_\alpha \geq s$ and (ii) when it is not. In scenario (i), $s'$ and $s''$ are non-positive, resulting in an infinite number of rounds since the target can never be reached. The average payoffs of PS and AU when playing against each other are thus $-c$ and 0, respectively (assuming that when a team's development speed is non-positive, its intermediate benefit, $b$, is zero). The condition for PS to be risk-dominant against AU

(see Eq 6 in Methods, and noting that the payoff of PS against another PS is the same as that of AS against another AS) reads

$$(1 - p_r)\left(\frac{sB}{2W} + \Pi_{22}\right) < \frac{B}{2W} + \Pi_{11} - c.$$

For sufficiently large $B$ (fixing $W$), this condition is reduced to, $p_r > 1 - 1/s$. That is, PS is risk-dominant against AU for the whole region (**II**), thereby ensuring that safe behaviour becomes promoted in that dilemma region.

Considering the second case in scenario (ii), where $s_\alpha < s$, the game is repeated for $\frac{W-s}{s-s_\alpha} + 1 = \frac{W-s_\alpha}{s-s_\alpha}$ rounds, which we denote here by $r$. Hence, the payoffs of PS and AU when playing with each other are given by, respectively

$$\frac{1}{r}\left(\pi_{12} + (r-1)\pi'_{12}\right),$$

$$\frac{p}{r}\left(B + \pi_{21} + (r-1)\pi'_{21}\right),$$

where

$$\pi'_{12} = \begin{cases} -c & \text{if } s > s_\alpha \geq 1 \\ -c + \dfrac{(1-s_\alpha)b}{s+1-2s_\alpha} & \text{if } s_\alpha < 1 \end{cases},$$

and

$$\pi'_{21} = \begin{cases} b & \text{if } s > s_\alpha \geq 1 \\ \dfrac{(s-s_\alpha)b}{s+1-2s_\alpha} & \text{if } s_\alpha < 1 \end{cases}.$$

Thus, for sufficiently large $B$, PS is risk dominant against AU when

$$p\frac{sB}{2W} + \frac{p}{r}B < \frac{B}{2W},$$

which is simplified to:

$$p_r > 1 - \frac{1}{s + 2Wr}. \tag{7}$$

This condition is easier to achieve for smaller $r$. Since $r$ is an increasing function of $s_\alpha$, to optimise the safety outcome, the highest possible $s_\alpha$ should be adopted, i.e. the strongest possible effort in slowing down the opponent should be made. Fig 4a shows the condition for different values of $s_\alpha$ in relation to $s$ (fixing the ratio $s_\alpha/s$). Numerical results in Fig 4b for a population of PS, AS and AU corroborate this analytical condition. Eq 7 splits the region (**II**) into two parts, (**IIa**) and (**IIb**), where PS is now also be preferred to AU in the first one. In part (**IIa**), the transition is stronger from AU to PS than vice versa (see Fig 3b). Recall that in the whole

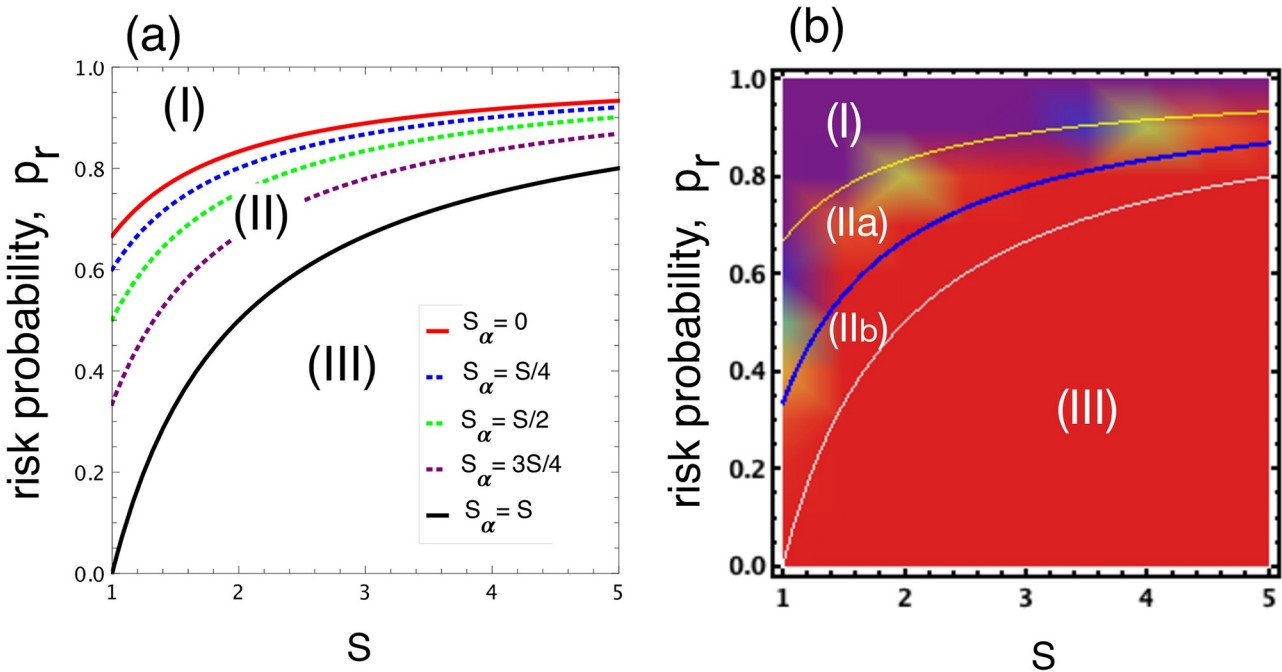

**Fig 4.** (a) Risk-dominant condition of PS against AU, as defined in Eq 7, for different ratio $s_\alpha/s$. The two solid lines correspond to when the ratio is 0 and 1, corresponding to the boundaries $p_r \in [1 - 1/s, 1 - 1/(3s)]$. The larger the ratio the smaller the Region (**II**) (between this line and the black line) is decreased, which disappears when $s_\alpha = s$. Panel (b): frequency of AU in a population of AS, AU, and PS (for $s_\alpha = 3s/4$). Region (**II**) is split into two (**IIa**) and (**IIb**) where PS is now also be preferred to AU in the first one. Parameters: $b = 4$, $c = 1$, $W = 100$, $B = 10000$, $\beta = 0.01$, $Z = 100$.

region (**II**) the transition is stronger from AS to AU, thus leading to a cyclic pattern between these three strategies.

When relaxing the assumption that $s_\beta = s_\alpha$ (see SI for the detailed calculation of payoffs), the effect of punishment for all variations of the parameters can be studied. The results are shown in Fig 5 (bottom row), for all the three regions shown in Fig 5 in inverse order. First, when looking at the right panel (bottom row) of Fig 5, one can observe that punishment does not alter the desired outcome (safety behaviour is the preferred outcome) in region (**I**), i.e. safe behaviour remains dominant. Significant less unsafe behaviour is observed in region (**II**), i.e. the middle panel (bottom row) of Fig 5, where it is not desirable, especially when $s_\alpha$ is small and $s_\beta$ is sufficiently large (purple area). However, punishment has an undesirable effect in region (**III**), i.e. the left panel (bottom row) of Fig 5, as it leads to reduction of AU when punishment is highly efficient (see the non-red area) while AU remains the preferred collective outcome in that region. The reason is that, for sufficiently small $s_\alpha$ and large $s_\beta$ (such that $s' > 0$ and $s' > s''$), PS gains significant advantage against AU, thereby dominating it even for low $p_r$.

In summary, reducing the development speed of unsafe players leads to a positive effect, especially when the personal cost is much less than the effect it induces on the unsafe player. Yet at the same time, it may lead to unwanted sanctioning effects in the region where risk-taking should be promoted.

## Reward vs punishment for promoting safety compliance

Here we investigate how positive incentives, as explained in Methods, influence the outcome in all three regions. The payoff matrix showing average payoffs among three strategies AS, AU

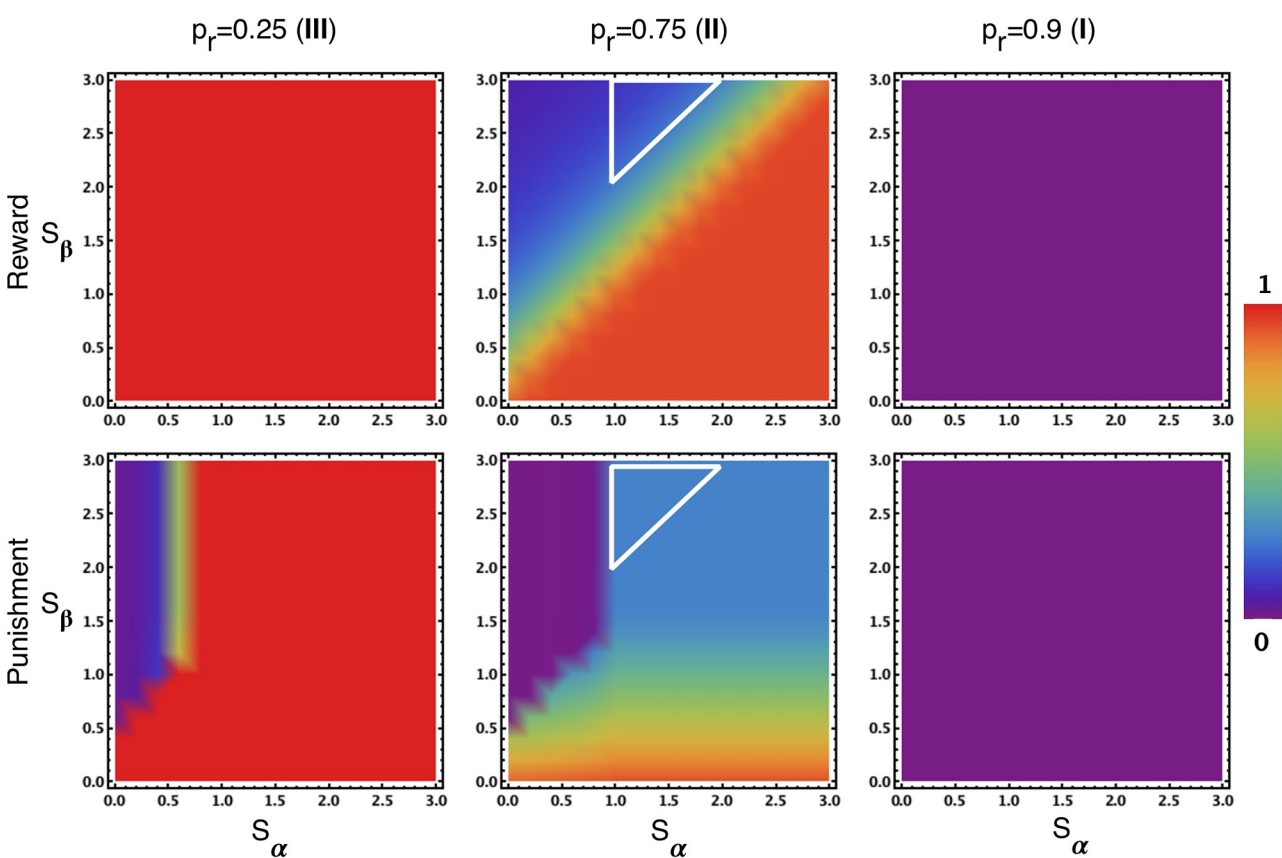

**Fig 5. AU Frequency: Reward (top row) vs punishment (bottom row) for varying $s_\alpha$ and $s_\beta$, for three regions.** In (**I**), both lead to no AU, as desired. In (**II**), punishment is more efficient except for when reward is rather costly but highly cost-efficient (the areas inside the white triangles). It is noteworthy that RS has very low frequency in all cases, as it catalyses the success of AS. In (**III**), RS always leads to the desired outcome of high AU frequency, while PS might lead to an undesired result of a reduced AU frequency (over-regulation) when highly efficient (non-red area). Parameters: $b = 4$, $c = 1$, $W = 100$, $B = 10000$, $s = 1.5$, $\beta = 0.01$, population size, $Z = 100$.

and RS reads

$$
\begin{array}{c}
\phantom{AS} \quad\quad AS \quad\quad\quad\quad AU \quad\quad\quad\quad RS \\
\begin{array}{c} AS \\ AU \\ RS \end{array}
\begin{pmatrix}
\frac{B}{2W} + \Pi_{11} & \Pi_{12} & \frac{B(1+s_\beta)}{W} + \Pi_{11} \\
p\left(\frac{sB}{W} + \Pi_{21}\right) & p\left(\frac{sB}{2W} + \Pi_{22}\right) & p\left(\frac{sB}{W} + \Pi_{21}\right) \\
\Pi_{11} & \Pi_{12} & \frac{B(1+s_\beta-s_\alpha)}{2W} + \Pi_{11}
\end{pmatrix}.
\end{array}
\tag{8}
$$

The payoff of RS against another RS is given under the assumption that reward is sufficiently cost-efficient, such that $1 + s_\beta > s_\alpha$; otherwise, this payoff would be $\Pi_{11}$. On the one hand, one can observe that RS is always dominated by AS. On the other hand, the condition for RS to be risk-dominant against AU is given by:

$$
p\left(\frac{sB}{2W} + \Pi_{22} + \frac{sB}{W} + \Pi_{21}\right) < \Pi_{12} + \frac{B(1 + s_\beta - s_\alpha)}{2W} + \Pi_{11},
$$

which, for sufficiently large $B$ (fixing $W$), is equivalent to

$$p_r > 1 - \frac{1 + s_\beta - s_\alpha}{3s}. \qquad (9)$$

Hence, RS can improve upon AS when playing against AU whenever $s_\beta > s_\alpha$ (recall that the condition for AS to be risk-dominant against AU is $p_r > 1 - 1/(3s)$). It is different from the peer punishment strategy PS that can lead to improvement even when $s_\beta \leq s_\alpha$.

Thus, under the above condition, a cyclic pattern emerges (see Fig 3b, considering that a neutral transition has arrows both ways): from AS to AU, to RS, then back to AS. In contrast to punishment, the rewarding strategy RS has a very low frequency in general (as it is always dominated by the non-rewarding safe player AS). Nonetheless, RS catalyses the emergence of safe behaviour.

Fig 5 (top row) shows the frequencies of AU in a population with AS and RS, for varying $s_\alpha$ and $s_\beta$, in comparison with those from the punishment model, for the three regions. One can observe that, in region (**II**), i.e. the middle panel (top row) of Fig 5, punishment is more (or at least as) efficient than reward in suppressing AU except for when incentivising is rather costly (i.e. sufficiently large $s_\alpha$) but highly cost-efficient ($s_\beta > s_\alpha$) (the areas inside the white triangles; see also S1 Fig for clearer difference with larger $\beta$). It is because only when incentive is highly cost-efficient, RS can take over AU effectively (see again Eq 9); and furthermore, the larger both $s_\alpha$ and $s_\beta$ are, the stronger the transition from RS to AS, to a degree that can overcome the transition from AS to AU. For an example satisfying these conditions, where $s_\alpha = 1.5$ and $s_\beta = 3.0$, see S4 Fig.

In regions (**I**) and (**III**), i.e. the right and left panels (top row) of Fig 5, similarly to punishment, the rewarding strategy does not change the outcomes, as is desired. Note however that differently from punishment, in region (**I**), i.e. the right panel (top row) of Fig 5, only AS dominates the population, while in the case of punishment, AS and PS are neutral and together dominate the population (see Fig 3, comparing panels c and f). Most interestingly, rewards do not harm region (**III**), i.e. the left panel (top row) of Fig 5, which suffers from over-regulation in the case of punishment because of the stronger transitions from RS to AS and AS to AU. Additional numerical analysis shows that all these observations are robust for larger $\beta$ (see S1 Fig).

In SI, we also consider the scenario where both peer reward and punishment are present, in a population of four strategies, AS, AU PS and RS (see S2 and S3 Figs). Since PS behaves in the same way as AS when interacting with RS, there is always a stronger transition from RS to PS. It results in an outcome in terms of AU frequency similar to the case when only PS is present, suggesting that, in a self-organized scenario, peer-punishment is more likely to prevail than peer-rewarding when individuals face a technological race.

Finally, it is noteworthy that all results obtained in this paper are robust if one considers that with some probability in each round UNSAFE players can be detected resulting in those UNSAFE players losing all payoff in that round [13]. This observation confirms the previous finding in a short-term AI regime that only participants' speeds matter (in relation to the disaster risk, $p_r$), and controlling the speeds is important to ensure a beneficial outcome (see also [13]).

## Discussion

In this paper we study the dynamics associated with technological races, those having the objective of being the first to bring some AI technology to market as a case study. The model proposed, however, is general enough for applicability to other innovation dynamics which

face the conflict between safety and rapid development [17, 23]. We address this problem resorting to a multiagent and complex systems approach, while adopting well established methods from evolutionary game theory and populations dynamics.

We propose a plausible adaptation of a baseline model [13] which can be useful when thinking about policies and regulations, namely incipient forms of community enforcing mechanisms, such as peer rewards and sanctions. We identify the conditions under which these incentives provide the desired effects while highlighting the importance of clarifying the risk disaster regimes and the time-scales associated with the problem. In particular, our results suggest that punishment—by forcibly reducing the development speed of unsafe participants—can generally reduce unsafe behaviour even when sanctions are not particularly efficient. In contrast, when punishment is highly efficient, it can lead to over-regulation and an undesired reduction of innovation, noting that a speedy and unsafe development is acceptable and more beneficial for the whole population whenever the risk for setbacks or disaster is low compared to the extra speed gained by ignoring safety precautions. Similarly, rewarding a safe co-player to speed up its development may, in some regimes, stimulate safe behaviours, whilst avoiding the detrimental impact of over-regulation.

These results show that, similarly to peer incentives in the context of one-shot social dilemmas (such as the Prisoner's Dilemma and the Public Goods Game) [31–40], strategies that target development speed in DSAIR can influence the evolutionary dynamics, but interestingly, they produce some very different effects from those of incentives in social dilemmas [41]. For example, we have shown that strong punishment, even when highly inefficient, can lead to improvement of safety outcome; while punishment in social dilemmas can promote cooperation only when highly cost-efficient. On the other hand, when punishment is too strong, it might lead to an undesired effect of over-regulation (reducing innovation where desirable), which is not generally the case in social dilemmas.

Incentives such as punishment and rewards have been shown to provide important mechanisms to promote the emergence of positive behaviour (such as cooperation and fairness) in the context of social dilemmas [31–40, 42, 43]. Incentives have also been successfully used for improving real world behaviours such as vaccination [44, 45]. Notwithstanding, all existing modelling approaches to AI governance [1, 13] do not study how incentives can be used to enhance safety compliance. Moreover, there have been incentive-modelling studies addressing other kinds of risk, such as climate change and nuclear war, see e.g. [37, 46, 47]. Following from an analysis of several large global catastrophic risks [20], it has been shown that the race for domain supremacy through AI and its related risks are rather unique. Analyses of climate change disasters primarily focus on participants' unwillingness to take upon themselves some personal cost for a desired collective target, and implies a collective risk for all parties involved [37]. In contrast, in a race to become leader in a particular AI application domain, the winner (s) will extract significant advantage relative to that of others. More importantly, this AI risk is also more directed towards individual developers or users than collective ones.

Our model and analysis of elementary forms of incentives thus provides an instrument for policy makers to ponder on the supporting mechanisms (e.g. positive and negative incentives), in the context of technological races [48–51]. Concretely, both sanctioning of wrong-doers (e.g. rogue or unsafe developers/teams) and rewarding of right-doers (e.g. safe-compliant developers/teams) can lead to enhancement of the desirable outcome (it being that of innovation or risk-taking in low risk cases, and safety-compliance in higher risk cases). Notably, while the former can be detrimental for innovation in low risk cases, it leads to a stronger enhancement for a wider range of effect-to-cost ratio of incentives. Thus, when it is not clear from the beginning what is the risk level associated (with the technology to be developed), then positive incentives appear to be the safer choice than negative ones (in line with historical

data on rewards usage in innovation policy in the UK [49] as well as suggestions for Covid-19 vaccine innovation policy [24]). This is the case for many kinds of technological races especially when data about the effect of a new technology is usually lacking and only becomes available when it has been created and used enough (see the Collingridge Dilemma [52]), as are the cases of the domain supremacy race through AI [21, 22] and the race for creating the first Covid-19 vaccines [24, 53]. On the other hand, when one can determine early on that the associated level of risk is sufficiently high (i.e. above a certain threshold as determined in our analysis), negative incentives might provide a stronger mechanism. For instance, high risk technologies such as new airplane models, medical products and biotech [54–56] might benefit from putting strong sanctioning mechanisms in place.

In the present modeling, we considered that development teams/players (adopting the same strategic behaviour) move at the same speed, similar to standard repeated games [16]. However, since these speeds can be very different especially when considering heterogeneity in teams' capacity (e.g. small/poor vs big/rich companies), we will need to consider a new time scale. There would be a possible time delay in players' decision-making, during the course of a repeated interaction, because they might want to wait for the outcome of a co-player's decision to see what choice he/she has adopted and/or will adopt in the next development round. Thus, a player has to decide whether to make an immediate move based on just present information —and hence be quicker to collect the next benefit and move faster in the race—but at the risk of making a worse choice, different from one that would have been chosen had the player already known the co-player's decision. Furthermore, counterfactual thinking might subsequently correct, in future choices, the choice made in the past—or delay its move to clarify the co-player's decision (thus, slower in collecting benefits and being slower in the race) [57]. Our future work aims to extend current repeated game models to capture this time delay aspect and study how it influences the outcomes of the repeated interactions. For instance, would reciprocal strategies such as tit-for-tat and win-stay-lose-shift [16, 58] still be successful, or would a new type of strategic behaviour emerge? Also, whether players should wait to see the co-player's move in due course, or should they make a move based on the present information? Moreover, since noise is a key factor driving the emergent strategic behaviours in the context of repeated games [16]—for instance when a team might (non-deliberately) make a mistake in the safety process, which might intensify the on-going race and trigger long-term retaliation—we will consider conflict resolution mechanisms such as apology and forgiveness [59–62] for simmering down the effects of noise on the race.

Additionally, the current model includes a binary-choice action choice (SAFE vs UNSAFE). As a generalisation of this binary-choice model we can consider continuous choice models where a player can choose the level of safety-precaution to adopt, where SAFE and UNSAFE correspond to the two extreme cases of complete precaution and no precaution at all, respectively. The player can also adjust the speed strategically during the race, e.g. depending on the current progress of other players and the stage of the race. This has been shown to be highly relevant in the context of climate change [63].

In short, our analysis has shown, within an idealised model of an AI race and using a game theoretical framework, that some simple forms of peer incentives, if used suitably (to avoid over-regulation, for example) can provide a way to escape the dilemma of acting safely even when speedy unsafe development is preferred. Future studies may look at more complex incentivising mechanisms [50] such as reputation and public image manipulation [64, 65], emotional motives of guilt and apology-forgiveness [60, 66], institutional and coordinated incentives [34, 46], and the subtle combination of different forms of incentive (e.g., stick-and-carrot approach and incentives for agreement compliance) [37, 39, 67–69].

## Appendix

### Details of analysis for three strategies AS, AU, CS

Let CS be a conditionally safe strategy, playing SAFE in the first round and choosing the same move as the co-player's choice in the previous round. We recall below the detailed calculations for this case, as described in [13], just for completeness. The average payoff matrix for the three strategies AS, AU, CS reads (for row player)

$$
\Pi = \begin{array}{c} AS \\ AU \\ CS \end{array}
\begin{array}{ccc}
AS & AU & CS
\end{array}
\left(
\begin{array}{ccc}
\frac{B}{2W} + \pi_{11} & \pi_{12} & \frac{B}{2W} + \pi_{11} \\
(1-p_r)\left(\frac{sB}{W} + \pi_{21}\right) & (1-p_r)\left(\frac{sB}{2W} + \pi_{22}\right) & (1-p_r)\left[\frac{sB}{W} + \frac{s}{W}\left(\pi_{21} + \left(\frac{W}{s}-1\right)\pi_{22}\right)\right] \\
\frac{B}{2W} + \pi_{11} & \frac{s}{W}\left(\pi_{12} + \left(\frac{W}{s}-1\right)\pi_{22}\right) & \frac{B}{2W} + \pi_{11}
\end{array}
\right). (10)
$$

The conditions (i) SAFE population has a larger average payoff than that of UNSAFE one, i.e. $\Pi_{AS,AS} > \Pi_{AU,AU}$, meaning by definition that a collective outcome is preferred and (ii) when is it the case that AS and CS are more likely to be imitated against AU (i.e., risk-dominant) will be derived below. First, for condition (i), it must hold that

$$
\frac{B}{2W} + \pi_{11} > (1-p_r)\left(\frac{sB}{2W} + \pi_{22}\right). \tag{11}
$$

Thus,

$$
p_r > 1 - \frac{B + 2W\pi_{11}}{sB + 2W\pi_{22}}, \tag{12}
$$

which is equivalent to (since $B/W \gg b$)

$$
p_r > 1 - \frac{1}{s}. \tag{13}
$$

This inequality means that, whenever the risk of a disaster or personal setback, $p_r$, is larger than the gain that can be gotten from a greater development speed, then the preferred collective action in the population is safety compliance.

Now, for deriving condition (ii), we apply the condition in Eq 6 (cf. Methods) to the payoff matrix $\Pi$ above,

$$
\frac{B}{2W} + \pi_{11} + \pi_{12} > (1-p_r)\left(\frac{3sB}{2W} + \pi_{21} + \pi_{22}\right). \tag{14}
$$

$$
\frac{s}{W}\left(\pi_{12} + \left(\frac{W}{s}-1\right)\pi_{22}\right) + \frac{B}{2W} + \pi_{11}
$$
$$
> (1-p_r)\left[\frac{sB}{2W} + \frac{sB}{W} + \frac{s}{W}\left(\pi_{21} + \left(\frac{W}{s}-1\right)\pi_{22}\right) + \pi_{22}\right], \tag{15}
$$

which are both equivalent to (since $B/W \gg b$)

$$
p_r > 1 - \frac{1}{3s}. \tag{16}
$$

The two boundary conditions for (i) and (ii), as given in Eqs 13 and 16, splits $s - p_r$ parameter space into three regions, as exhibited in Fig 6a:

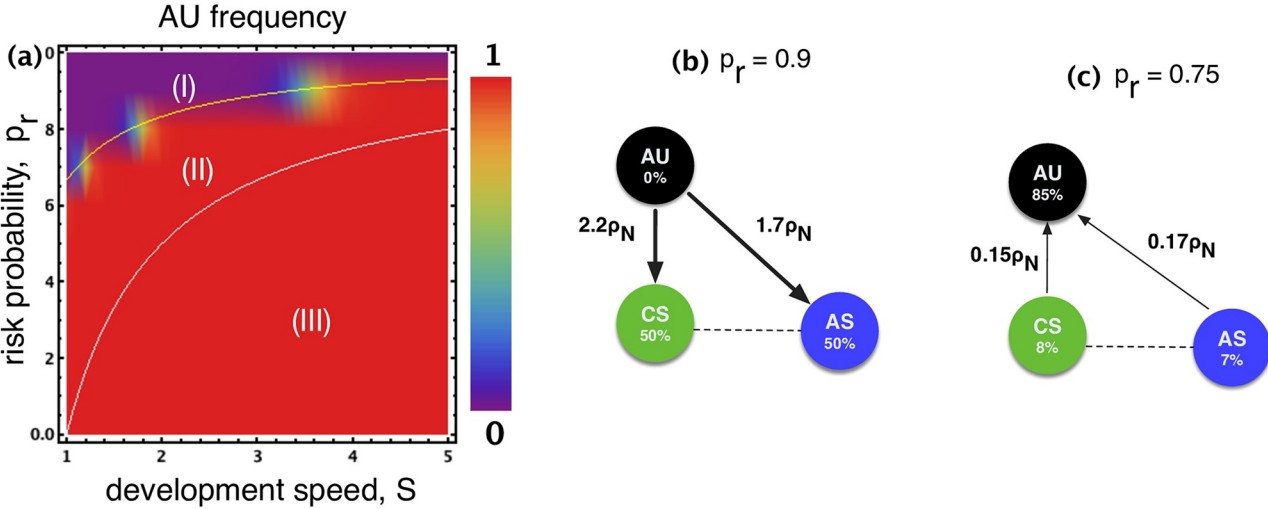

**Fig 6.** Panel **(a)** as in Fig 1 in the main text, added here for ease of following. Panels **(b)** and **(c)** show the transition probabilities and stationary distribution (see Methods). In panel (c) AU dominates, corresponding to region (**II**), whilst in panel (b) AS and CS dominate, corresponding to region (**I**). For a clear presentation, we indicate just the stronger directions. Parameters: $b = 4$, $c = 1$, $W = 100$, $B = 10^4$, $Z = 100$, $\beta = 0.1$; In panel **(b)** $p_r = 0.9$; in panel (c) $p_r = 0.6$; in both (b) and (c) $s = 1.5$.

(I). when $p_r > 1 - \frac{1}{3s}$: This corresponds to the *AIS compliance zone*, in which safe AI compliance is both preferred collectively and that unconditionally (AS) and conditionally (CS) safe development is the social norm (an example for $s = 1.5$ is given in Fig 6b: $p_r > 0.78$);

(II). when $1 - \frac{1}{3s} > p_r > 1 - \frac{1}{s}$: This intermediate zone is the one that captures a dilemma because, collectively, safe AI developments are preferred, though the social dynamics pushes the whole population to the state where all develop AI in an unsafe manner. We shall refer to this zone as the *AIS dilemma zone* (for $s = 1.5$, $0.78 > p_r > 0.33$, see Fig 6c);

(III). when $p_r < 1 - \frac{1}{s}$: This defines the *AIS innovation zone*, in which unsafe development is not only the preferred collective outcome but also the one the social dynamics selects.

It is noteworthy in an early DSAI, only two parameters $s$ and $p_r$ are relevant. Intuitively, when $B/W$ is sufficiently large, the average payoff obtained from winning the race (i.e. gaining $B$) is significantly larger than the intermediate benefit a player can obtain from each round of the game (at most $b$), making the latter irrelevant. Thus, the only way to improve a player's average payoff (i.e. individual fitness) is to increase the player's speed of gaining $B$. On the other hand, AU's payoff is scaled by a factor $(1 - p_r)$.

## Calculation for $\pi_{PS,AU}$ and $\pi_{AU,PS}$ in general case

Below $R$ denotes the average number of rounds; $B_1$ and $B_2$ the benefits PS and AU might obtain from the winning benefit $B$ when either of them wins the race by being the first to have made $W$ development steps; $b_1$ and $b_2$ the intermediate benefits PS and AU might obtain in each round of the game; $p_{loss}$ is the probability that all the benefit is not lost when AU wins and draws the race; Clearly, all these values depend on the development speeds ($s'$ for PS and

$s''$ for AU).

$$\pi_{PS \ vs \ AU} = \frac{1}{R(s', s'')} \left[ \pi_{12} + B_1(s', s'') + (R(s', s'') - 1)(-c + b_1(s', s'')) \right]$$

$$\pi_{PS \ vs \ AU} = p_{loss}(s', s'') \times \frac{1}{R(s', s'')} \left[ \pi_{21} + B_2(s', s'') + (R(s', s'') - 1) b_2(s', s'') \right]$$

where

$$B_1(s', s'') = \begin{cases} B & \text{if } s' > 0 \ \& \ s'' \le 0 \\ B & \text{if } s' > 0 \ \& \ \frac{W-s}{s''} > \frac{W-1}{s'} \\ B/2 & \text{if } s' > 0 \ \& \ \frac{W-s}{s''} = \frac{W-1}{s'} \\ 0 & \text{otherwise} \end{cases}$$

$$B_2(s', s'') = \begin{cases} B & \text{if } s' \le 0 \ \& \ s'' > 0 \\ B & \text{if } s'' > 0 \ \& \ \frac{W-s}{s''} < \frac{W-1}{s'} \\ B/2 & \text{if } s'' > 0 \ \& \ \frac{W-s}{s''} = \frac{W-1}{s'} \\ 0 & \text{otherwise} \end{cases}$$

$$b_1(s', s'') = \begin{cases} (1 - p_{fo}) \frac{s'b}{s'+s''} + p_{fo} b & \text{if } s' > 0 \ \& \ s'' > 0 \\ b & \text{if } s' > 0 \ \& \ s'' \le 0 \\ 0 & \text{otherwise} \end{cases}$$

$$b_2(s', s'') = \begin{cases} (1 - p_{fo}) \frac{s''b}{s'+s''} & \text{if } s' > 0 \ \& \ s'' > 0 \\ (1 - p_{fo}) b & \text{if } s' \le 0 \ \& \ s'' > 0 \\ 0 & \text{otherwise} \end{cases}$$

$$R(s', s'') = \begin{cases} +\infty & \text{if } s' \le 0 \ \& \ s'' \le 0 \\ \frac{W-1}{s'} + 1 & \text{if } s' > 0 \ \& \ s'' \le 0 \\ \frac{W-s}{s''} + 1 & \text{if } s' \le 0 \ \& \ s'' > 0 \\ 1 + \min\left\{ \frac{W-s}{s''}, \frac{W-1}{s'} \right\} & \text{otherwise} \end{cases}$$

$$p_{loss}(s', s'') = \begin{cases} p(= 1 - p_r) & \text{if } s'' > 0 \ \& \ \frac{W-s}{s''} \le \frac{W-1}{s'} \\ 1 & \text{otherwise} \end{cases}$$

## Supporting information

**S1 Fig. AU Frequency: Reward (top row) vs punishment (bottom row) for varying $s_\alpha$ and $s_\beta$, for three regions, for stronger intensity of selection ($\beta = 0.1$).** Other parameters are the same as in Fig 5 in the main text. The observations in that figure is also robust for larger intensities of selection.
(TIF)

**S2 Fig. Transitions and stationary distributions in a population of four strategies AU, AS, PS and RS, for three regions.** Only stronger transitions are shown for clarity. Dashed lines denote neutral transitions. In addition, note that PS is equivalent to AS when interacting with PS, i.e. there is always a stronger transition from RS to PS than vice versa. Parameters as in Fig 2.
(TIF)

**S3 Fig. AU frequency for varying $s_\alpha$ and $s_\beta$, in a population of four strategies AS, AU, PS and RS, for three regions.** The outcomes in all regions are similar to the case of punishment (without reward) in Fig 5. The reason is that there is always a stronger transition from RS to PS than vice versa. Parameters as in Fig 5.
(TIF)

**S4 Fig. Transitions and stationary distributions in a population of three strategies AU, AS, with either PS (top row) or RS (bottom row), in region (II) ($pr = 0.75$): Left column ($\beta = 0.01$), right column ($\beta = 0.1$).** The parameters of incentives fall in the white triangles in Fig 5 and S1 Fig: $s_\alpha = 1.5$, $s_\beta = 3$. We observe that the frequency of AU is lower in case of reward than that of punishment. Other parameters as in Fig 2.
(TIF)

**S1 Data.**
(NB)

## Author Contributions

**Conceptualization:** The Anh Han, Luís Moniz Pereira, Tom Lenaerts, Francisco C. Santos.

**Formal analysis:** The Anh Han, Luís Moniz Pereira, Tom Lenaerts, Francisco C. Santos.

**Funding acquisition:** The Anh Han, Luís Moniz Pereira, Tom Lenaerts, Francisco C. Santos.

**Investigation:** The Anh Han, Luís Moniz Pereira, Tom Lenaerts, Francisco C. Santos.

**Methodology:** The Anh Han, Luís Moniz Pereira, Tom Lenaerts, Francisco C. Santos.

**Software:** The Anh Han.

**Validation:** The Anh Han, Luís Moniz Pereira, Tom Lenaerts, Francisco C. Santos.

**Visualization:** The Anh Han, Luís Moniz Pereira, Tom Lenaerts, Francisco C. Santos.

**Writing – original draft:** The Anh Han, Luís Moniz Pereira, Tom Lenaerts, Francisco C. Santos.

**Writing – review & editing:** The Anh Han, Luís Moniz Pereira, Tom Lenaerts, Francisco C. Santos.

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
