## [Decision Letter · Decision Letter 0]

16 Sep 2020

PONE-D-20-16449

Mediating artificial intelligence developments through negative and positive incentives

PLOS ONE

Dear Dr. Han,

Thank you for submitting your manuscript to PLOS ONE. After careful consideration, we feel that it has merit but does not fully meet PLOS ONE’s publication criteria as it currently stands. Therefore, we invite you to submit a revised version of the manuscript that addresses the points raised during the review process.

We look forward to receiving your revised manuscript.

Kind regards,

Alberto Antonioni, PhD

Academic Editor

PLOS ONE

Journal Requirements:

3. Please ensure that you refer to Figure 8 adn 9 in your text as, if accepted, production will need this reference to link the reader to the figure.

Additional Editor Comments (if provided):

The authors should address reviewers' constructive comments before considering the paper for publication in PLoS ONE. Overall, reviewers appreciated the work but give some useful suggestions to improve the presentation of the paper.

Reviewers' comments:

Reviewer's Responses to Questions

**Comments to the Author**

1. Is the manuscript technically sound, and do the data support the conclusions?

Reviewer #1: Partly

Reviewer #2: Yes

Reviewer #3: Partly

2. Has the statistical analysis been performed appropriately and rigorously? 

Reviewer #1: N/A

Reviewer #2: N/A

Reviewer #3: N/A

3. Have the authors made all data underlying the findings in their manuscript fully available?

Reviewer #1: Yes

Reviewer #2: Yes

Reviewer #3: Yes

4. Is the manuscript presented in an intelligible fashion and written in standard English?

Reviewer #1: Yes

Reviewer #2: Yes

Reviewer #3: Yes

5. Review Comments to the Author

Reviewer #1: In the manuscript, the authors revealed the effects of rewards and punishments on AI developments (or other technology development races). They analytically and numerically solved the dynamics among some strategies (AS, AU, PS, and RS) by the means of Evolutionary Game Theory (EGT) and found that both rewards and punishments are effective for the developments.

I think the most contribution of this manuscript and Ref. 13 is that the authors invented the payoff matrix for the AI developments based on game theory. The payoff matrix is different from the one in social dilemma games, which EGT has frequently been applied. Although the original matrix was invented in Ref. 13 by the same authors, the authors included new strategies related to rewards and punishments in this manuscript. Thus, they successfully extended the study of this AI development games.

I recommend the publication of this manuscript in PLoS ONE after the following technical points are improved.

There are two points I find the authors can improve.

1. Organization of the manuscript. I think the authors can rearrange the structure of the manuscript to be better read.

Examples are below.

- The sixth and seventh paragraphs (lines 46-71) in the introduction can be combined with the fifth paragraph (lines 134-142) in the Materials and methods.

- When I read the model, especially Eq. 2., I forget what B and W are. Those variables are defined in the sixth paragraph (lines 46-57) in the Introduction, which is far from the model section. Perhaps, the Related Work section can be moved to the earlier part of the Introduction?

In short, if the authors could restructure the Introduction, Related Work, and Model, it would be beneficial to readers for better understanding.

2. Numerical calculations. It is better to give the detail procedures of the numerical calculation. I roughly understood how the analytical solutions were derived. However, I don't know how the numerical one is obtained. For example, in Figs. 2 and 4, the white lines are analytically obtained by calculating the fixation probabilities of strategies. A fixation probability represents the PROBABILITY that a single mutant replaces the existing population. However, the color maps in Figs. 2 and 4 obtained by the numerical calculations represents the FREQUENCY of the strategies. I don't know how the frequency is calculated. Did you numerically get the frequency by iterating Eq. 4 until the stationary distributions are obtained? If the authors give the detail procedure of the numerical calculations as a subsection, it would be great.

The followings are minor points.

- Could you add an intuitive explanation that why the risk-dominance thresholds are obtained only by two variables, p_r and s? I mathematically understand, if B/W is large enough, some variables can be ignored, but I don't know how it can be interpreted?

- Please add the clear definition of B. Lines 47 and 53 give the relation to that point but is not clearly defined. Does B imply the net benefit of the AI achievement?

- Line 315. "dynamics ." The space should be removed.

Reviewer #2: This is a very well written paper with a clear and concise model. My only suggestion would be to potentially enlarge the discussion. I believe the incentives discussed probably also apply to fields other than AI. Vaccine development or biotech in general, comes to mind.

I strongly recommend this paper is published.

Reviewer #3: This paper examines an evolutionary game theoretic model of a technological innovation race to study the effects of peer rewarding/punishment mechanisms. The technical part (mathematical modeling and analysis) seems sound and well presented, which should be worth publication in PLOS ONE.

Having said that, I think there are a few major issues in framing and presentation, which make the proposed model rather unjustified as a model of AI development. Details are explained below. The authors are strongly recommended to make a major revision in framing, presentation and justification of the model and its assumptions.

1. It was not clearly justified why the AI development is a particularly good interpretation of this model.

The model is a generic evolutionary game theoretic (EGT) model that represents competition among multiple players trying to reach a goal faster than the competitors while rewarding/penalizing the competitors according to their strategies. While this model setting itself is reasonably constructed and of some interest to the EGT community, it does not appear to describe the AI development scenario in particular. Actual AI technology development companies do not engage in a series of discrete pairwise game play events with their competitors (especially given that their development stages are asynchronous due to different development speeds), and they certainly would not have any incentives to impose the proposed peer rewarding/punishment (especially rewarding) on their opponents in the real industry ecosystem. These mismatches with reality make the model questionable as a model of AI development. There may be a better analogy for interpretation of this model (e.g., rapid vaccine development in which multiple competitors are also cooperating, to some extent, to achieve a global public health goal, for example).

2. The assumptions made about the speed of development do not seem to adequately capture the reality.

The key parameter of interest in this study is the speed of technology development, on which several assumptions were made: (1) The faster the speed is, the more unsafe the developed technology will be. (2) The speed of UNSAFE development is a fixed universal constant, and each developer has no control to adjust that speed, but is only able to choose either SAFE (slow) or UNSAFE (fast). (3) The competitor developer *can* change the opponent's speed by peer rewarding/punishment (even though it can't change its own speed in a similar way). None of these assumptions seems well justified, especially in the context of AI technology development. Unlike medical and pharmaceutical development that requires extensive clinical testing, technology development in computational domains can happen sometimes at a very rapid pace and result in a breakthrough, not necessarily in a risky technology. The speed of development is most likely not a GO-NOGO binary choice but should be more gradual strategic parameter that each developer can adjust by itself. Meanwhile, it would be rather difficult to influence the competitor's development speed from the outside, aside from filing lawsuits on, e.g., IP infringements (or perhaps by cyber-attacks).

Other miscellaneous points:

* Figure 2 was referred to earlier than Figure 1. The order of those figures should be reversed.

* It would be better to re-introduce definitions of symbols (S, B, W, p_r, etc.) in the Materials and Methods section.

* In page 8: "a cyclic pattern emerges (see Figure 3b)": Figure 3b does not show a cycle.

* In page 9: "see Figure 5, comparing panels c and f": Figure 5 does not have such panels.

* In page 9: "This observation confirms the observation": A redundant and unclear expression.

* Figure 1 does not have orange or blue areas.

* The direction of the blue line in Figure 1 does not look correct. Shouldn't its slope be negative, given that the rewarder reduces her speed?

6. PLOS authors have the option to publish the peer review history of their article (what does this mean?). If published, this will include your full peer review and any attached files.

Reviewer #1: No

Reviewer #2: No

Reviewer #3: No

---

## [Author Response · Author response to Decision Letter 0]

27 Nov 2020

We have attached a PDF with response to reviewers

---

## [Decision Letter · Decision Letter 1]

14 Dec 2020

Mediating artificial intelligence developments through negative and positive incentives

PONE-D-20-16449R1

Dear Dr. Han,

We’re pleased to inform you that your manuscript has been judged scientifically suitable for publication and will be formally accepted for publication once it meets all outstanding technical requirements.

Kind regards,

Alberto Antonioni, PhD

Academic Editor

PLOS ONE

Additional Editor Comments (optional):

The current version of the manuscript can be considered for publication on PLoS ONE as positively evaluated by the reviewers.

Reviewers' comments:

Reviewer's Responses to Questions

**Comments to the Author**

1. If the authors have adequately addressed your comments raised in a previous round of review and you feel that this manuscript is now acceptable for publication, you may indicate that here to bypass the “Comments to the Author” section, enter your conflict of interest statement in the “Confidential to Editor” section, and submit your "Accept" recommendation.

Reviewer #1: All comments have been addressed

Reviewer #3: All comments have been addressed

2. Is the manuscript technically sound, and do the data support the conclusions?

Reviewer #1: Yes

Reviewer #3: Yes

3. Has the statistical analysis been performed appropriately and rigorously? 

Reviewer #1: Yes

Reviewer #3: Yes

4. Have the authors made all data underlying the findings in their manuscript fully available?

Reviewer #1: Yes

Reviewer #3: Yes

5. Is the manuscript presented in an intelligible fashion and written in standard English?

Reviewer #1: Yes

Reviewer #3: Yes

6. Review Comments to the Author

Reviewer #1: (No Response)

Reviewer #3: The authors have addressed my comments (at least the limitations were discussed in the Discussions section). I recommend this manuscript to be published in PLOS ONE.

7. PLOS authors have the option to publish the peer review history of their article (what does this mean?). If published, this will include your full peer review and any attached files.

Reviewer #1: **Yes: **Genki Ichinose

Reviewer #3: No

---

## [Editor Report · Acceptance letter]

8 Jan 2021

PONE-D-20-16449R1 

Mediating artificial intelligence developments through negative and positive incentives

Dear Dr. Han:

I'm pleased to inform you that your manuscript has been deemed suitable for publication in PLOS ONE. Congratulations! Your manuscript is now with our production department. 

Kind regards, 

on behalf of

Dr. Alberto Antonioni 

Academic Editor

PLOS ONE